# Nested Melanoma and Superficial Spreading Melanoma with Prominent Nests—A Retrospective Study on Clinical Characteristics and PRAME Expression

**DOI:** 10.3390/diagnostics15172279

**Published:** 2025-09-08

**Authors:** Daniela Lenders, Valentin Aebischer, Maximilian Gassenmaier, Matthias Hahn, Gisela Metzler, Stephan Forchhammer

**Affiliations:** 1Department of Dermatology, University of Tuebingen, 72076 Tuebingen, Germany; daniela.lenders@med.uni-tuebingen.de (D.L.);; 2Dermatopathology Friedrichshafen, 88048 Friedrichshafen, Germany

**Keywords:** melanoma, nested melanoma, superficial spreading melanoma, dysplastic nevi, PRAME

## Abstract

**Background:** Nested melanoma is a rare subtype of superficial spreading melanoma. Due to its typical histology, characterized by the predominance of large melanocytic nests in an extensive horizontal spread, it is challenging to distinguish it from other benign nested melanocytic lesions. There is a need to identify additional histopathological parameters that can support the diagnosis of nested melanoma. **Methods:** In this retrospective case-control study, we analyzed immunohistochemical staining for PRAME in 10 cases of superficial spreading melanoma with prominent nests and 26 nested melanomas. Dysplastic melanocytic nevi were used as the control group. **Results:** We found a diffuse PRAME positivity (>75% of melanocytes) in 60% of superficial spreading melanoma with prominent nests and 19% of nested melanoma cases, whereas the control group showed no diffuse PRAME positivity. Furthermore, using Melan A immunohistochemistry, we found an absence of pagetoid spread in 31% of nested melanoma and single cells in suprabasal epidermal layers in 69% of cases. All cases with no pagetoid spread were PRAME negative, whereas 28% of cases with a mild pagetoid spread demonstrated diffuse PRAME positivity. **Conclusions:** We found lower PRAME positivity in nested melanoma compared to superficial spreading melanoma with prominent nests. Particularly in cases without pagetoid intraepidermal spread of melanocytes, negative PRAME staining does not rule out the possibility of nested melanoma. The diagnosis should be made based on typical histomorphological and clinical criteria.

## 1. Introduction

The histological diagnosis of melanoma is still challenging due to the morphological diversity and numerous histological variants. Generally, the most common histological subtypes include superficial spreading melanoma, nodular melanoma, acral lentiginous melanoma, and lentigo maligna melanoma. In 2012, Kutzner et al. described a variant of superficial spreading melanoma called “melanomas composed exclusively or predominantly of large nests (MLN)” or nested melanoma [1]. This melanoma subtype is characterized by an arrangement of melanocytes in large, rounded “cannonball” junctional nests of varying shapes and sizes. As other commonly described melanoma criteria are often not identified, such as a pagetoid intraepidermal spread of melanocytes, diagnosing MLN can be challenging. It is critical to distinguish it from other nested melanocytic lesions, such as benign melanocytic nevi or dysplastic nevi. Clinically, nested melanomas exhibit typical melanoma features, such as asymmetry in color and shape, multiple colors, irregular borders, and a size greater than 6 mm. In addition to the typical dermatoscopic melanoma criteria, nested melanomas present quite specifically with large, irregular round to oval dots and globules, which correspond to the large nests seen in the histological sections [1]. Epidemiological data on nested melanoma are scarce, as only small series have been reported. It appears to predominantly affect older individuals, typically in the sixth to eighth decade of life [1,2]. Reported tumor thickness is generally low, and the prognosis has been considered favorable compared to conventional superficial spreading melanoma [2,3].

PRAME is an immunohistochemical marker that has become increasingly important in the routine diagnosis of melanoma in recent years. PRAME (PReferentially expressed Antigen in Melanoma) is a melanoma-associated antigen that was first isolated by autologous T-cells in a patient with metastatic melanoma [4]. It is a member of the cancer testis antigen (CTA) family and is expressed in the testis, placenta, adrenals, endometrium, and sebaceous gland, but not in other normal tissues [5]. PRAME is overexpressed in various tumors, such as lung cancer, breast cancer, and ovarian cancer [6,7,8]. PRAME expression is also a common event in melanoma and was previously investigated for treatment purposes [9]. Lezcano et al. analyzed PRAME immunohistochemistry for the histological differentiation of melanomas and benign melanocytic lesions [10], showing a diffuse PRAME expression in 87% of metastatic melanomas and 83.2% of primary melanomas. In comparison, negativity for PRAME was found in 86.4% of the analyzed melanocytic nevi. Numerous studies were recently conducted on the diagnostic use of PRAME in adult melanomas, which have confirmed these results but mostly found lower expression levels in melanoma [11,12,13,14].

In this retrospective case-control study, we compared 10 cases of SSM with prominent nest, 26 cases of nested melanoma, and 36 dysplastic nevi in terms of their immunohistochemical PRAME and Melan A staining. The aim was to determine whether PRAME is a reliable immunohistochemical marker for distinguishing between nested melanoma and benign nevi. Additionally, we analyzed the clinical course of patients with nested melanoma, as there are still limited clinical data available for this rare melanoma entity. This study provides one of the first systematic analyses of PRAME expression and clinical features in nested melanoma, compared to superficial spreading melanoma with prominent nests and dysplastic nevi.

## 2. Materials and Methods

### 2.1. Database Search and Study Cohort

A database search of the Archive for Dermatopathology at the University Hospital of Tübingen was conducted for the period from 2014 to 2024 using NEXUS PAS.NET Version 1.22.26.23095. The keywords were “nested” and “nested type melanoma”. The initial diagnosis was made by at least one board-certified dermatopathologist based on the presence of an extensive horizontal spread of large irregular melanocytic nests of varying sizes and shapes with a tendency to confluence and moderate to marked cytological atypia, usually in sun-damaged skin, as well as clinical information [1,15]. The initial diagnosis was primarily based on H&E-stained sections. In some cases, other immunohistochemical stains were also used for diagnosis, such as Melan A, Hmb45, Ki67, and, in some recent cases, PRAME. Additionally, we received three more cases of diagnosed nested melanomas as part of a collaboration with Dermatohistopathology Friedrichshafen. The total cohort included 47 patients. Of these, two were excluded because there was no additional tumor material available. Four patients were excluded due to diagnostic uncertainty, as they were not clearly diagnosed as melanoma but as borderline tumors, such as superficial atypical melanocytic proliferation of uncertain significance (SAMPUS). Five other patients were excluded as they were diagnosed as having melanoma in situ and not invasive melanoma. All diagnoses were reviewed by three board-certified dermatopathologists (SF, MH, GM), and the diagnosis of melanoma was confirmed in all cases included. This resulted in a final count of 36 patients with invasive melanoma who were included in our study.

### 2.2. Clinical Assessment

The following clinical parameters were retrospectively collected from electronic patient records (SAP i.s.h med system, version 2024) from the university hospital Tuebingen: age, sex, the localization of the primary tumor, the presence of ulceration, nevus association, regression, Breslow tumor thickness/T stage, Clark level, tumor stage (AJCC 2017), duration of follow-up, presence of recurrence, and metastasis or additional melanomas.

### 2.3. Immunohistochemical Analysis

FFPE tissue samples were processed using the standard protocols for diagnostic histological examination. Sections were cut using a rotary microtome (Microm HM355S, Thermo Fisher Scientific, Walldorf, Germany) and placed on poly-L-lysine-coated microscopy slides (catalog number J2800AMNZ, Menzel-Gläser, Thermo Fisher Scientific, Braunschweig, Germany). Immunohistochemistry (IHC) was performed with an antibody to PRAME (dilution of 1:25, clone QR005, DCS, Hamburg, Germany) on an automated immunostainer (Leica Bond-MAX, Leica Biosystems, Wetzlar, Germany) using a red chromogen (FastRed). The percentage of PRAME immunoreactive cells was scored as previously described [10] (0: no staining at all; 1+: 1–25%; 2+: 26–50%; 3+: 51–75%; 4+: >75% positive cells). PRAME staining of any intensity in >75% of epidermal or >75% of dermal melanocytes was interpreted as diffuse. Sebaceous glands on the examined slides and PRAME-positive melanoma were used as the positive control. PRAME expression was independently quantified by two investigators (DL and SF) in the epidermal and dermal melanocytic components. Furthermore, we analyzed staining for Melan A (dilution 1:150, clone A103, Dako, Jena, Germany) regarding a pagetoid intraepidermal spread of melanocytes. Melan A immunostaining was evaluated on full-slide sections. Pagetoid spread was quantified semi-quantitatively by assessing the presence of suprabasal melanocytes across the entire epidermis. No restriction to a predefined high-power field or absolute cell number was applied; rather, the extent of suprabasal melanocytes was categorized as absent, focal (1+), or extensive (2+).

The analysis was conducted without knowledge of the histological diagnosis and the clinical course. In the event that the two investigators obtained a different classification of PRAME expression or pagetoid infiltration, the specimens were jointly examined under the microscope to reach a consensus.

Since nested melanomas, by definition, do not exhibit marked pagetoid spread, cases in which a distinct suprabasal distribution of melanocytes was observed upon evaluation in a Melan A staining were reclassified as superficial spreading melanomas (SSM) with prominent nests. Cases with absent or only minimal pagetoid infiltration were classified as bona fide nested melanoma.

Dysplastic melanocytic compound nevi were used as the control group. For this purpose, the cases were matched based on age and, when possible, gender.

A flowchart of the cohort selection process, including inclusion and exclusion criteria as well as histological and immunohistochemical review, is shown in Appendix A.

### 2.4. Statistics

Statistical analyses were performed using SPSS (IBM Corp., Armonk, NY, USA; Version 28.0.0.0) and Microsoft Excel (Microsoft Corporation, Redmond, WA, USA; Version 16.0, Build 17932.20428). One-way ANOVA was used to compare PRAME expression across the three groups (0, 1+, 2+). Descriptive statistics and simple group comparisons were conducted in Excel. Post hoc analyses following ANOVA were performed using Fisher’s exact test. In addition, subgroup analyses according to sex (Fisher’s exact test) and age (Mann–Whitney U-test) were carried out. A *p*-value < 0.05 was considered statistically significant.

### 2.5. Ethics Statement

This study was approved by the Institutional Review Board of the University of Tübingen (Project ID: 855/2023BO2, 8 January 2024) and was conducted in accordance with the Declaration of Helsinki.

## 3. Results

### 3.1. Epidemiological Data

Included in our study were 36 patients with an initial histological diagnosis of nested melanoma. Upon re-evaluating the pagetoid spread, 10 tumors were reclassified as SSM with prominent nests, while 26 tumors were classified as bona fide nested melanoma. The epidemiological data showed that both groups were generally comparable, but with some differences. While the median age for SSM with prominent nests was 77.1 years, patients with nested melanoma had a median age of 60.5 years. In the SSM group, the majority of patients were female (70%), whereas nested melanoma in our cohort predominantly occurred in male patients (77%). The tumors in both groups were primarily located on the trunk. Both groups consisted exclusively of low-risk tumors, with a median tumor thickness of 0.56 mm in the SSM group and 0.4 mm in the nested melanoma group. Neither group showed ulcerated tumors. As a result, all tumors in the SSM group were stage Ia, while in the nested melanoma group, 25 tumors were classified as stage Ia and 1 tumor as stage Ib according to AJCC 2017. Follow-up data was available for only one patient in the SSM group and 18 patients in the nested melanoma group. In these patients, no events such as recurrence, metastasis, or death due to the disease occurred during the follow-up period (median of 4.9 years for nested melanoma). Despite the limited cohort size, both melanoma subgroups were comparable with respect to tumor thickness and stage. However, patients with superficial spreading melanoma with prominent nests were older and more frequently female compared to those with nested melanoma. All epidemiological data are summarized in Table 1.

### 3.2. Comparison of PRAME and Melan A Immunohistochemistry in SSM with Prominent Nests, Nested Melanoma and Dysplastic Melanocytic Nevi

We analyzed PRAME and Melan A expression regarding the nuclear PRAME expression pattern and pagetoid spread of melanocytes by immunohistochemical staining in SSM with prominent nests, nested melanoma, and dysplastic melanocytic nevi. The immunohistochemical results of each case are summarized in Table 2. In total, 8 of 10 (80%) SSM with prominent nests and 13 of 26 (50%) nested melanomas were found to have PRAME expression in the junctional and/or dermal compartment. In 5 cases of SSM with prominent nests and in 10 cases of nested melanoma, the positivity for PRAME was stronger in the junctional compartment than the dermal compartment. In the other positive cases, PRAME was equally expressed in the junctional and dermal compartments. Diffuse PRAME positivity (staining of >75% of epidermal or >75% of dermal melanocytes; 4+) was found in 6 of 10 (60%) SSM with prominent nests and in 5 of 26 (19%) nested melanomas (see Figure 1). Subgroup analyses according to sex and age did not reveal statistically significant differences in diffuse PRAME expression. Diffuse PRAME expression was observed in 46.2% of female patients (6/13) and 21.7% of male patients (5/23; *p* = 0.15). Median age was 66 years in diffuse PRAME-positive and 70 years in diffuse PRAME-negative patients (*p* = 0.88). Furthermore, we analyzed the expression of Melan A regarding pagetoid intraepidermal spread of melanocytes. Ten cases with marked pagetoid spreading melanocytes were classified as SSM with prominent nests. Eight of 26 (31%) bona fide nested melanomas showed no pagetoid spreading of melanocytes, and 18 of 26 (69%) nested melanomas demonstrated single melanocytes in higher epidermal layers.

The control group of dysplastic melanocytic nevi revealed PRAME expression in 9 of 36 cases (25%) (an exemplary case can be seen in Appendix A). In all cases, positivity was stronger in the junctional compartment than the dermal compartment, and diffuse PRAME positivity was not found. In 5 of 36 cases (14%), we identified single melanocytes in higher epidermal layers with Melan A immunostaining. Strong pagetoid spread was not detectable in any of the control cases (see Table 2).

Considering the pagetoid spread of melanocytes in Melan A immunostaining, the analyzed melanomas can be divided into three groups: 8 cases showed no pagetoid spread (0; subgroup 1; see Figure 2a–c), 18 cases showed focal suprabasal melanocytes (+1; subgroup 2; see Figure 2d–f), and 10 cases showed extensive pagetoid spread (+2; subgroup 3, re-classified as SSM with prominent nests; see Figure 2g–i). When analyzing PRAME expression across the three subgroups (no pagetoid spread, 1+, and 2+), one-way ANOVA revealed significant overall differences (F = 4.47, *p* = 0.019). Post-hoc testing showed a significant difference between cases without pagetoid spread (0/8 diffuse PRAME-positive) and those with marked pagetoid spread (2+: 6/10 diffuse PRAME-positive; *p* = 0.013). Comparisons between the no pagetoid and 1+ groups (5/18 diffuse PRAME-positive; *p* = 0.281) and between the 1+ and 2+ groups (*p* = 0.125) were not statistically significant (see Figure 3).

## 4. Discussion

Regarding the epidemiological data, we found that our cohort of nested melanoma predominantly consisted of older patients with an average age of 60.5 years at diagnosis. This is consistent with the data in the literature, where nested melanoma is primarily described in older patients and has therefore been referred to by Longo et al. as “nested melanoma of the elderly” [2]. Nevertheless, we also detected nested melanoma in younger patients, and our youngest patient was 41 years old. Most nested melanomas were found on the trunk and extremities, as also described by Kutzner et al. [1]. Our data indicate that all cases exhibited low tumor thickness, no ulceration, and a low T stage. No metastases were found, so all patients were classified as stage I (AJCC2017). In the follow-up, no distant metastasis or tumor recurrence was detected in any of these patients. These data raise the question of whether these cases are truly melanomas or a distinct type of atypic melanocytic nevus. In particular, tumors without pagetoid spread and with absent PRAME expression may in fact represent atypical or dysplastic melanocytic nevi rather than true melanomas. This issue has previously been addressed in the literature. Kutzner et al. analyzed the genomic aberrations of these tumors using array comparative genomic hybridization (aCGH) and were able to detect multiple gains and losses of several chromosomes or chromosomal parts [1]. The same pattern of chromosomal aberrations is well described as being characteristic of conventional superficial spreading melanoma [16,17]. In addition, fluorescence in situ hybridization (FISH) showed melanoma-typical aberrations for the genes *RREB1*, *MYB*, and *CCND1*, whereas the control group of melanocytic nevi did not show aberrations in aCGH and FISH [1]. In addition, Pennacchia et al. found the *BRAFV600E* mutation in three of eight nested melanoma cases [3]. These abnormalities can indicate evidence of malignancy and support the possible biological relationship between nested melanoma and superficial spreading melanoma. Since the Breslow tumor thickness holds the greatest prognostic value [18], most thin melanomas of other subtypes also did not show distant metastases, and the average survival rate for a tumor thickness < 1 mm is 85–90% [18,19,20]. However, even thin melanomas can metastasize in rare cases [21], making early detection crucial.

Nested melanoma is a rare melanoma entity whose distinction from benign nested lesions, such as melanocytic nevi, can pose histological challenges, as their histopathological picture does not match well with common criteria for melanoma [1,19]. For this reason, we investigated the role of PRAME in differentiating between SSM with prominent nests, nested melanomas, and benign dysplastic melanocytic nevi. The utility of PRAME in diagnosing melanoma is widely discussed in the literature based on the description by Lezcano et al. that a diffuse PRAME expression is highly specific for non-spindle cell melanomas [10]. They described a diffuse positivity for PRAME in around 90% of conventional melanomas but only in 0.7% of nevi. A limitation of Lezcano et al.’s initial study is that only thicker melanomas (mean Breslow thickness 3.3 mm) were examined.

In our study, we found PRAME positivity in 50% of nested melanomas, whereas diffuse positivity, considered specific for melanomas, was found in only 19%. Thus, we found lower PRAME positivity in nested melanoma than in other melanoma subtypes. In line with previous descriptions, nested melanoma was characterized by compact intraepidermal nests and only limited pagetoid spread [1,2,3]. Our results extend these observations by showing that diffuse PRAME expression was uncommon in this group, underscoring the limited sensitivity of PRAME for the recognition of nested melanoma. This is in contrast to the situation in conventional SSM and may reflect the borderline nature of some lesions. In contrast, superficial spreading melanomas with prominent nests showed PRAME positivity in 60% of cases, which is in keeping with previous reports on PRAME expression in conventional SSM [10,11]. These data emphasize that, despite their nested growth pattern, such cases behave immunohistochemically like typical SSM and should be classified accordingly. Additionally, we examined Melan A immunohistochemistry in our cohort to investigate pagetoid spread of melanocytes in the epidermis, as the existing literature reports that this highly specific melanoma criterion is often absent in nested melanoma [1]. In our study, we found isolated cells in higher epidermal layers in most cases of nested melanoma, which was also described by Pennacchia et al. [3]. Surprisingly, we detected strong pagetoid infiltration in 10 of 36 (28%) cases initially diagnosed as nested melanoma, which were therefore reclassified as SSM. Correlating the pagetoid intraepidermal spread of melanocytes with PRAME expression revealed that more pronounced pagetoid spread was associated with a higher number of cases showing diffuse PRAME expression. It is also possible that nested melanoma and superficial spreading melanoma represent a continuum of the same entity, with pagetoid spread and PRAME expression increasing over the course of the disease, which would also align with the similar genetic alterations observed in these two entities. Both types of melanomas occur in moderately chronically photodamaged skin, most frequently in the trunk. Morphologically, both entities show a wide-spreading lesion with predominantly horizontal tumor growth without a prominent lentiginous growth pattern. Our data suggest that, especially in cases without pronounced pagetoid spread, PRAME is not a suitable marker to diagnose nested melanoma, as none of the cases in this group exhibited diffuse PRAME positivity. In some cases, however, PRAME positivity can assist the pathologist by providing additional support for the correct diagnosis of nested melanoma in distinguishing it from dysplastic nevus, as diffuse PRAME positivity was not found in any cases in the control group. Since only thin melanomas were present in our cohort, it is comparable to a previous study by Gassenmaier et al., who investigated PRAME positivity in thin melanomas with respect to their metastatic potential [11]. They found diffuse PRAME staining in 58.6% of cases, which did not serve as a marker for distinguishing metastasizing from non-metastasizing thin melanomas. This rate of PRAME positivity is nearly identical to the rate of diffuse positive PRAME expression observed in SSM with prominent nests of 60% in our cohort and indicates a substantially lower rate of diffuse PRAME positivity in nested melanomas compared to other melanoma subtypes with similar risk stratification. Interestingly, diffuse PRAME positivity is more frequently reported in melanomas in chronically sun-damaged skin [10] and in older patients [22] compared to cases with low cumulative UV damage. However, assuming that nested melanoma primarily occurs in older individuals, these findings were not reproducible in our study. Table 3 provides a comparative overview of the clinical, histopathological, and immunohistochemical features of nested melanoma versus melanocytic nevi, supplemented by the findings of our cohort. This side-by-side summary highlights the diagnostic challenges of nested melanoma and underscores the limited sensitivity of PRAME in this context. Importantly, the table also emphasizes the value of combining histomorphology, clinical information, and immunohistochemistry for accurate diagnosis. As such, Table 3 may serve as a practical reference for dermatopathologists confronted with this diagnostically challenging entity.

Our study has several limitations. The relatively small sample size represents an important limitation of this study. This is, however, largely inevitable due to the rarity of nested melanoma as a diagnostic entity. Consequently, the results should be interpreted with caution, and confirmation in a larger multicenter series would be desirable. Another limitation is that the cohort does not include melanomas with a high tumor thickness and melanoma-specific events such as recurrence, metastasis, and death. Since the diagnostic criteria for nested melanoma are imprecise and there is no clinical correlation with unfavorable follow-up, it cannot be ruled out with absolute certainty that some of the melanomas in our cohort are in fact well-nested atypic nevi, which could explain the lower PRAME expression. A molecular analysis of the tumors is not available, as this is not routinely performed for nested melanomas. The retrospective setting of this study is also a limitation, which may lead to a selection bias of the groups and thus affect the generalizability of the results. Another limitation is the routine use of PRAME in the diagnosis of some melanocytic tumors at our institute since the year 2020, which may have had an impact on the study design. This could have led to a higher rate of PRAME-positive nested melanomas, as it cannot be ruled out that tumors with negative PRAME expression were classified as nevi.

## 5. Conclusions

In summary, while diffuse PRAME expression is found in some cases of nested melanoma, it does not provide sufficient diagnostic assistance, particularly in cases without pagetoid intraepidermal spread of melanocytes. Our data suggests that some of the lesions diagnosed as nested melanoma—particularly those lacking pagetoid spread and PRAME expression—may be better classified as atypical nevi. The diagnosis of nested melanoma should always be made in conjunction with typical histomorphological and clinical criteria, specifically a broad, extremely well-nested melanocytic lesion along with a history of a progressive growth in an older patient. Generally, a multidisciplinary approach involving close collaboration between the treating clinician and the reporting dermatopathologist is crucial to make an adequate diagnosis [23,24].

Nevertheless, we recommend further studies with larger sample sizes, although this is challenging due to the rarity of this melanoma subtype.

**Table 3 diagnostics-15-02279-t003:** Comparison between nested melanoma and melanocytic nevi.

Characteristics		Nested Melanoma	Melanocytic Nevi
Clinical findings	Patient history	- Progressive growth in size - Changes in color	- Typically no significant increase in size in adult patients - No changes in color
	Age	- Older patients (mostly patients > 65 years, and rarely in younger patients) [2] This study: - Nested melanoma: mean age 60.5 years - SSM with prominent nests: mean age 77.1 years	- In older patients, new nevi should generally no longer appear and should be primarily considered suspicious
	Macroscopic findings	- Typical melanoma features: asymmetry of color and shape, multiple colors, irregular borders, size > 6 mm	- Symmetric, well-defined melanocytic lesion, uniform in color, size of <5 mm
	Dermatoscopy	- Typical common melanoma features: atypical pigment network, blue structures, signs of regression, atypical vascular patterns - Characteristics: large, irregular round to oval dots and globules [2]	- Uniform, well-defined pigment network, often round or oval structures, homogenous coloration, absence of irregular streaks or signs of regression
Histopathological findings	Morphology	- Broad melanocytic lesion - Extremely well nested: melanocytes in large, rounded “cannonball” junctional nests of varying shapes and sizes - Moderate to severe cellular atypia [1]	- Melanocytic nevus: regular arrangement of melanocytes in nests or cords at dermoepidermal junction, uniform nevus cells - Dysplastic nevus: more irregular asymmetrical melanocytic nests, mild cellular atypia, bridging of melanocytes
	Immunohistochemistry	- Usually no or mild pagetoid intraepidermal spread of melanocytes in Melan A staining - More pronounced pagetoid spread suggests transition to superficial spreading melanoma (SSM) type This study: - Nested melanoma: Diffuse PRAME expression in 19% - SSM with prominent nests: Diffuse PRAME expression in 60%	- Usually no or mild pagetoid intraepidermal spread of melanocytes in Melan A staining - Usually no diffuse PRAME expression, cases with diffuse PRAME positivity should be critically assessed [11]
	Genetic aberrations	- Array CGH: multiple gains and losses on several chromosomes - FISH: melanoma-typical aberrations, e.g., *RREB1*, *MYB*, and *CCND1* - *BRAFV600* mutation possible but not well studied [3]	- No aberrations in aCGH and FISH - Frequent *BRAFV600* mutation (60–80%) [25,26]

## Figures and Tables

**Figure 1 diagnostics-15-02279-f001:**
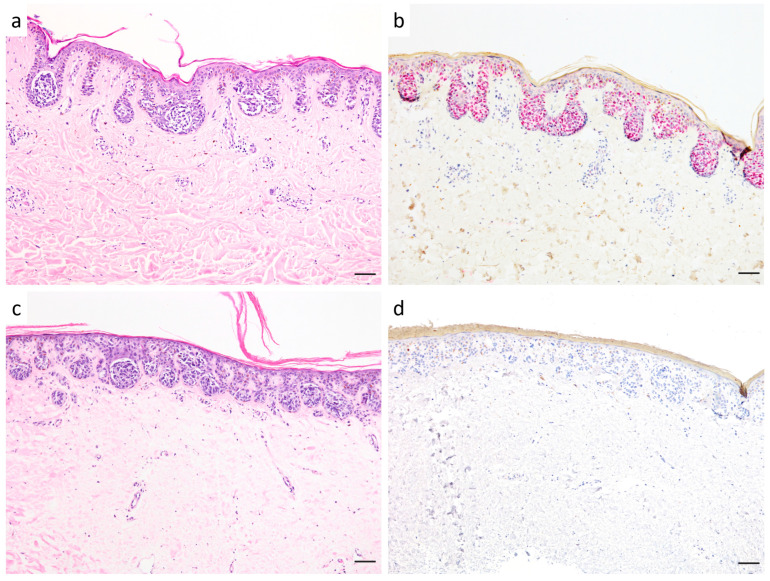
(**a**,**b**) (patient no. 31): nested melanoma with large junctional nests of nevoid melanocytes in “clod” pattern (**a**). PRAME is expressed in >75% of melanocytes, defined as diffuse expression pattern (**b**). (**c**,**d**) (patient no. 25): typical architecture of nested melanoma (**c**) with negativity for PRAME (**d**). Scale bar is 100 µm.

**Figure 2 diagnostics-15-02279-f002:**
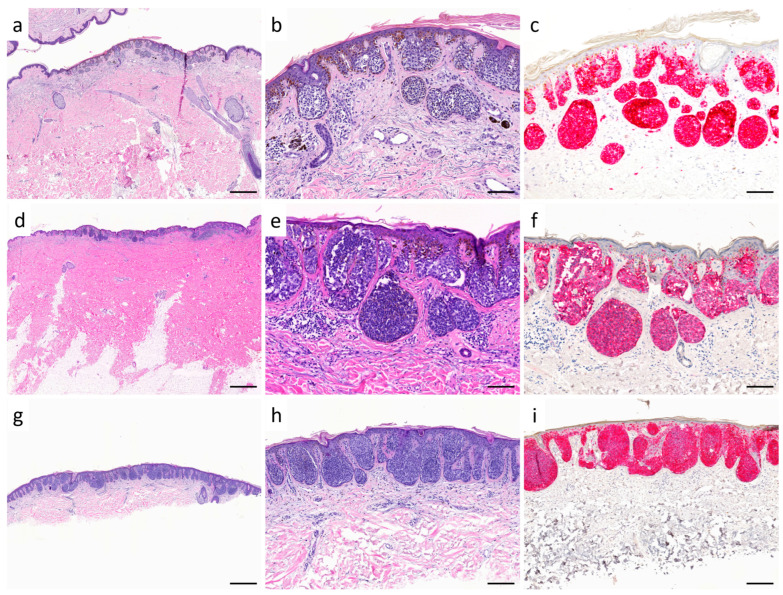
Pagetoid intraepidermal spread of melanocytes in SSM with prominent nests and nested melanoma. (**a**–**c**) (patient no. 27): typical architecture of nested melanoma in Hematoxylin–Eosin (HE) staining (scale bar: (**a**), 500 µm; (**b**), 100 µm). This case shows no melanocytes in suprabasal epidermal layers in Melan A immunohistochemistry (scale bar: (**c**), 100 µm). (**d**–**f**) (patient no. 35) highlight expansive, well-nested lesion (scale bar: (**d**), 1 mm; (**e**), 100 µm), with few melanocytes in higher epidermal layers (classified as 1+) in Melan A immunohistochemistry ((**f**): scale bar is 100 µm). (**g**,**h**) (patient no. 29) was initially classified as nested melanoma based on morphological criteria (scale bar: (**g**), 500 µm; (**h**), 200 µm). Pronounced pagetoid spread (classified as 2+) is only detectable through Melan A immunostaining ((**i**): scale bar is 200 µm) and therefore reclassified as SSM with prominent nests.

**Figure 3 diagnostics-15-02279-f003:**
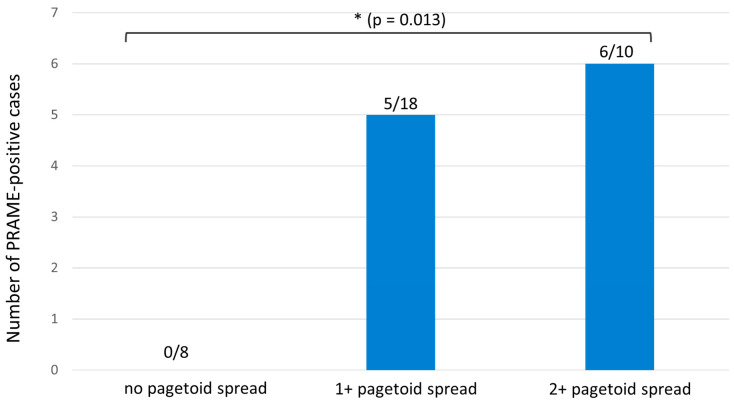
Diffuse PRAME expression in nested melanoma subgroups according to pagetoid spread (absolute numbers). One-way ANOVA revealed significant overall differences (F = 4.47, *p* = 0.019); post-hoc analysis showed significance between the 0 and 2+ groups (*p* = 0.013). The asterisk (*) indicates statistical significance (*p* < 0.05).

**Table 1 diagnostics-15-02279-t001:** Epidemiological data.

Characteristics	SSM with Prominent Nests	Nested Melanoma
Total, n	10	26
Age, years, mean, (SD; range)	77.1 (10.6, 59–86)	60.5 (12.7, 41–84)
Sex, n (%)		
Male	3 (30)	20 (77)
Female	7 (70)	6 (23)
**Localization of primary tumor, n (%)**		
Trunk	6 (60)	20 (77)
Head/neck	0 (0)	1 (4)
Upper extremities	1 (10)	3 (11)
Lower extremities	3 (30)	2 (8)
Ulceration, n (%)		
Yes	0 (0)	0 (0)
No	10 (100)	26 (100)
Regression, n (%)		
Yes	1 (10)	1 (4)
No	10 (90)	24 (96)
Nevus association, n (%)		
Yes	2 (20)	4 (15)
No	8 (80)	22 (85)
Clark Level, n (%)		
I	0 (0)	0 (0)
II	4 (40)	13 (50)
III	5 (50)	9 (35)
IV	1 (10)	4 (15)
V	0 (0)	0 (0)
Breslow tumor thickness, mm, median (range)	0.56 (0.3–0.72)	0.4 (0.25–1.2)
Breslow tumor thickness/T stage (TNM classification, AJCC 2017), number of patients (%)		
≤1.0 mm/T1	10 (100)	25 (96)
>1.0–2.0 mm/T2	0 (0)	1 (4)
>2.0–4.0 mm/T3	0 (0)	0 (0)
>4.0 mm/T4	0 (0)	0 (0)
Melanoma stage (8th edition, AJCC 2017)		
Ia, n (%)	10 (100)	25 (96)
Ib, n (%)	0 (0)	1 (4)
II, n (%)	0 (0)	0 (0)
III, n (%)	0 (0)	0 (0)
IV, n (%)	0 (0)	0 (0)
Patients with follow-up data, n	1	18
Duration of follow-up, years, mean (SD; range)	1 (0; 1)	4.9 (3.2; 0–12)
Recurrences, n (%)	0 (0)	0 (0)
Metastases, n (%)	0 (0)	0 (0)
Multiple melanomas, n (%)	2 (20)	4 (15)

**Table 2 diagnostics-15-02279-t002:** Comparison of PRAME and Melan A immunohistochemistry in SSM with prominent nests, nested melanomas and dysplastic melanocytic nevi. Abbreviations: F, female; M, male. Tumors with “pagetoid spread” 0 and 1+ are classified as bona fide nested melanoma, tumors with “pagetoid spread” 2+ were reclassified as SSM with prominent nests.

SSM with Prominent Nests and Nested Melanoma						Dysplastic Melanocytic Nevi					
Patient No./Sex	Age (y)	PRAME Junctional	PRAME Dermal	Diffuse PRAME	Pagetoid Spread	Patient No./Sex	Age (y)	PRAME Junctional	PRAME Dermal	Diffuse PRAME	Pagetoid Spread
1/M	49	0	0	−	1+	37/M	44	0	0	−	0
2/M	76	1+	0	−	0	38/M	71	0	0	−	0
3/M	70	1+	0	−	1+	39/M	51	0	0	−	0
4/M	52	2+	0	−	1+	40/M	73	0	0	−	0
5/F	66	4+	1+	+	2+	41/M	62	0	0	−	0
6/M	70	0	0	−	1+	42/M	70	0	0	−	0
7/M	77	0	0	−	1+	43/M	78	0	0	−	0
8/M	59	0	0	−	2+	44/M	52	0	0	−	0
9/F	41	0	0	−	0	45/F	45	0	0	−	0
10/F	64	4+	3+	+	2+	46/F	68	0	0	−	0
11/F	63	2+	1+	−	0	47/M	60	0	0	−	0
12/F	62	0	0	−	2+	48/F	63	3+	2+	−	1+
13/F	71	4+	4+	+	2+	49/M	75	2+	0	−	0
14/F	58	0	0	−	1+	50/F	54	0	0	−	0
15/M	77	0	0	−	0	51/M	78	0	0	−	0
16/M	84	0	0	−	1+	52/M	81	0	0	−	0
17/M	59	0	0	−	0	53/M	51	0	0	−	0
18/F	72	4+	4+	+	2+	54/F	76	0	0	−	0
19/M	76	2+	0	−	2+	55/M	79	1+	0	−	1+
20/M	63	0	0	−	1+	56/M	64	0	0	−	0
21/F	71	1+	0	−	1+	57/F	69	0	0	−	0
22/F	66	4+	3+	+	2+	58/F	63	1+	0	−	1+
23/F	79	2+	0	−	2+	59/F	76	0	0	−	1+
24/M	81	0	0	−	1+	60/M	81	0	0	−	0
25/F	81	0	0	−	0	61/F	86	0	0	−	0
26/M	57	3+	2+	−	1+	62/M	53	0	0	−	0
27/M	73	0	0	−	0	63/M	71	3+	0	−	0
28/M	79	0	0	−	0	64/M	77	0	0	−	0
29/M	86	4+	4+	+	2+	65/M	80	1+	0	−	0
30/M	58	4+	2+	+	1+	66/M	59	0	0	−	0
31/M	60	4+	0	+	1+	67/M	60	0	0	−	0
32/M	84	4+	4+	+	1+	68/M	83	3+	2+	−	1+
33/F	46	4+	4+	+	1+	69/F	44	0	0	−	0
34/M	52	3+	2+	−	1+	70/M	55	1+	0	−	0
35/M	69	1+	0	−	1+	71/M	68	0	0	−	0
36/M	74	4+	4+	+	1+	72/M	72	3+	0	−	0

## Data Availability

The data sets analyzed during the current study are available from the corresponding author upon reasonable request.

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
