# Peer review of "Nested Melanoma and Superficial Spreading Melanoma with Prominent Nests—A Retrospective Study on Clinical Characteristics and PRAME Expression"

_diagnostics, 2025, doi:10.3390/diagnostics15172279_

Round 1

Reviewer 1 Report

Comments and Suggestions for Authors

The authors analysed 36 invasive melanomas that were initially histologically diagnosed as nested melanomas. After a systematic morphological and immunohistochemical analysis, the authors reclassified the cases in the group of bona fide nested melanomas (n=26), in which there was no significant suprabasal pagetoid spread, and in the group of superficial spreading melanomas with prominent nests (n=10). They analysed PRAME expression in both groups and in additional 36 dysplastic naevi as a control group.

PRAME was diffusely positive in 60% of 10 superficial spreading melanomas (which is in accordance with other studies of superficial spreading melanomas), in 28% of 18 nested melanomas with some pagetoid spread, no of 8 nested melanoma without pagetoid spread and no dysplastic naevi.

The article is generally well written and the results is very relevant for diagnostic of melanocytic tumours. The authors concluded that nested melanomas are less frequently PRAME positive than superficial spreading melananomas and the diagnostic utility of PRAME may be lower in recognizing this particular type of melanomas.

The major question is whether nested melanomas are true melanomas or could be better classified as dysplastic naevi. As nicely shown, there is a spectrum from purely nested lesions to superficial speeding melanomas regarding pagetoid spread. The authors correctly discussed limitations of their study and mentioned that some of the lesions diagnosed as nested melanomas may not be true melanomas but rather dysplastic naevi.

I suggest following modifications:

  1. Based on clinical context and morphology, “nested melanomas” are part of the spectrum of superficial spreading melanomas. Lower rate of PRAME expression in so called nested melanomas therefore probably means that many of the lesions diagnosed as nested melanomas, especially those without pagetoid spread are not melanomas, as indicated by PRAME expression. The authors should more clearly state this possibility in the discussion. No, more emphasis is made explaining PRAME is not sensitive in recognition of nested melanomas.
  2. I would suggest to modify the title: Nested melanoma and superficial spreading melanoma with prominent nests—a retrospective study on clinical characteristics and PRAME expression

Author Response

Reviewer 1 Comments to the Author

The authors analysed 36 invasive melanomas that were initially histologically diagnosed as nested melanomas. After a systematic morphological and immunohistochemical analysis, the authors reclassified the cases in the group of bona fide nested melanomas (n=26), in which there was no significant suprabasal pagetoid spread, and in the group of superficial spreading melanomas with prominent nests (n=10). They analysed PRAME expression in both groups and in additional 36 dysplastic naevi as a control group.

PRAME was diffusely positive in 60% of 10 superficial spreading melanomas (which is in accordance with other studies of superficial spreading melanomas), in 28% of 18 nested melanomas with some pagetoid spread, no of 8 nested melanoma without pagetoid spread and no dysplastic naevi.

The article is generally well written and the results is very relevant for diagnostic of melanocytic tumours. The authors concluded that nested melanomas are less frequently PRAME positive than superficial spreading melananomas and the diagnostic utility of PRAME may be lower in recognizing this particular type of melanomas.

The major question is whether nested melanomas are true melanomas or could be better classified as dysplastic naevi. As nicely shown, there is a spectrum from purely nested lesions to superficial speeding melanomas regarding pagetoid spread. The authors correctly discussed limitations of their study and mentioned that some of the lesions diagnosed as nested melanomas may not be true melanomas but rather dysplastic naevi.

I suggest following modifications:

Comment 1: Based on clinical context and morphology, “nested melanomas” are part of the spectrum of superficial spreading melanomas. Lower rate of PRAME expression in so called nested melanomas therefore probably means that many of the lesions diagnosed as nested melanomas, especially those without pagetoid spread are not melanomas, as indicated by PRAME expression. The authors should more clearly state this possibility in the discussion. No, more emphasis is made explaining PRAME is not sensitive in recognition of nested melanomas.

Response 1: We thank the reviewer for this important comment. We agree that the possibility that some tumors diagnosed as nested melanomas, particularly those without pagetoid spread, may in fact represent atypical nevi should be stated more explicitly. We have therefore revised the Discussion and Conclusion to emphasize that negative PRAME expression in the absence of pagetoid spread supports the notion that some of these lesions may not represent true melanomas, and PRAME has limited sensitivity in the recognition of such cases. (page 9, lines 250-252; page 11, lines 350-352)

Comment 2: I would suggest to modify the title: Nested melanoma and superficial spreading melanoma with prominent nests—a retrospective study on clinical characteristics and PRAME expression

Response 2: We have modified the title of the manuscript accordingly.

Reviewer 2 Report

Comments and Suggestions for Authors

Abstract:

Add to the background that it is an early subtype/variant for SSM.

Introduction

Add an epidemiological context for this entity. What is known about the frequency/incidence, clinical and prognosis?

Reformulate the purpose in such a way as to potentiate the study as well as increase the attractiveness to readers.

Material and method

Add a flow chart for the selection criteria. Add inclusion and exclusion criteria in the study.

Add an area or a certain number of cells for the quantification of Melan A.

Add the manufacturer and state of the statistical program used.

Results

Given that the batch is numerically low, did you evaluate its homogeneity?

Line 158: compartment

Can some comparisons of expressions associated with sex/age be completed?

Discussions

Paragraph 209-213 can be eliminated

Comparisons with the specialized literature should be performed for the two categories separately, it would show a more complete picture of the pathology.

Indeed, a major problem of the study, marked in the limitations, consists in the small group.

The conclusion should be written separately.

You can add two columns to table 3 with the results for a comparative analysis between the study carried out and the literature.

Author Response

Reviewer 2 Comments to the Author

Abstract:

Comment 1:  Add to the background that it is an early subtype/variant for SSM.

Response 1: We thank the reviewer for this suggestion. We have revised the Abstract and now state that nested melanoma is considered subtype of superficial spreading melanoma. (page 1, lines 10-11)

Introduction

Comment 2:  Add an epidemiological context for this entity. What is known about the frequency/incidence, clinical and prognosis?

Response 2: In the Introduction we now include an epidemiological context, summarizing that nested melanoma is rare, primarily described in elderly patients, with limited data on incidence and prognosis, but generally associated with low tumor thickness and favorable outcomes. (page 1-2, lines 40-45; page 2 lines 48-52)

Comment 3:  Reformulate the purpose in such a way as to potentiate the study as well as increase the attractiveness to readers.

Response 3: We agree with the reviewer and have reformulated the final paragraph of the Introduction to emphasize the novelty and relevance of our study, underlining that it is one of the few systematic analyses of PRAME expression and clinical features in nested melanoma compared to SSM with prominent nests and dysplastic nevi. (page 2, lines 74-76)

Material and method

Comment 4:  Add a flow chart for the selection criteria. Add inclusion and exclusion criteria in the study.

Response 4: A flow chart illustrating patient selection, including inclusion and exclusion criteria, has been added as Supplementary Figure 1.

Comment 5:  Add an area or a certain number of cells for the quantification of Melan A.

Response 5: We have clarified that Melan A evaluation was performed on full-slide assessment and that pagetoid spread was quantified based on the presence of suprabasal melanocytes in the entire epidermis, not limited to a specific field or cell count. (page 3, lines 120-124)

Comment 6:  Add the manufacturer and state of the statistical program used.

Response 6: We have revised the Methods section (2.4, Statistics) accordingly and specified the statistical programs and versions used. (page 4, lines 140-147)

Results

Comment 7: Given that the batch is numerically low, did you evaluate its homogeneity?

Response 7: We thank the reviewer for this remark. We have clarified in the Results that, despite the limited cohort size, both groups were comparable regarding tumor thickness and stage. At the same time, we highlight that there were differences in age and sex distribution, with patients with superficial spreading melanoma being older and more often female compared to those with nested melanoma. (page 4, lines 169-172)

Comment 8: Line 158: compartment

Response 8: We thank the reviewer for noting this typographical error. It has been corrected from “compartmen” to “compartment.” (page 5, line 182)

Comment 9: Can some comparisons of expressions associated with sex/age be completed?

Response 9: Subgroup analyses according to sex and age were performed and added in the results section. (pages 5-6, lines 188-192)

Discussions

Comment 10: Paragraph 209-213 can be eliminated

Response 10: We agree with the reviewer and have removed the respective paragraph from the Discussion to avoid redundancy.

Comment 11: Comparisons with the specialized literature should be performed for the two categories separately, it would show a more complete picture of the pathology.

Response 11: We have expanded the Discussion by comparing nested melanoma and SSM with prominent nests separately to relevant reports in the literature, thereby providing a more comprehensive overview. (page 10, lines 278-289)

Comment 12:  Indeed, a major problem of the study, marked in the limitations, consists in the small group.

Response 12: We agree with the reviewer. We have emphasized the limitation of the small cohort size more clearly in the Discussion and explained that this reflects the rarity of the entity. (page 11, line 329-332)

Comment 13: The conclusion should be written separately.

Response 13: The Conclusions have now been separated into a distinct section at the end of the manuscript. (page 11, lines 347-359)

Comment 14:  You can add two columns to table 3 with the results for a comparative analysis between the study carried out and the literature.

Response 14: We thank the reviewer for this constructive idea. To highlight the added value of our study without overloading the table, we have directly incorporated the findings of our cohort into Table 3 at the relevant positions (e.g., age, histopathology, PRAME expression) and added additional references where appropriate. This allows for a clear side-by-side comparison with the published literature.

Reviewer 3 Report

Comments and Suggestions for Authors

After reading the manuscript my major concerns are as follows:

  1. Figure 3 and statistical analysis of data: How is it possible to compare (statistically) some values to the control group, whose nominative value is zero??? The Figure 3 should be corrected.

    Please, present numbers of “pagetoid spread cases”, instead of “cases in %”

  2. Since there are 3 groups (no pagetoid, 1+, and 2+) the one-way ANOVA should be applied. The Student's t-test is restricted to comparison based on only 2 groups, not 3. Besides, how to compare statistically some values to the control group, whose value has no cases (with no pagetoid spread)?

  3. Please, correct p-value statistics on the Figure 3. The decimal system in English is based on a period, not a comma.

  4. Table 3 is very important as future directions and needs more explanation in the Discussion.

Author Response

Reviewer 3 Comments to the Author

Comment 1:  Figure 3 and statistical analysis of data: How is it possible to compare (statistically) some values to the control group, whose nominative value is zero??? The Figure 3 should be corrected. Please, present numbers of “pagetoid spread cases”, instead of “cases in %”

Response 1: We thank the reviewer for this valuable comment. Figure 3 has been revised to display the absolute numbers of diffusely PRAME-positive cases per subgroup (0/8, 5/18, 6/10) instead of percentages, providing a clearer representation of the data. The group without pagetoid spread is still shown for completeness, but a direct statistical comparison to a “zero-value” control has been omitted, as this does not yield meaningful information. Statistical testing was instead performed across all three subgroups using ANOVA (see Response 2).

Comment 2:  Since there are 3 groups (no pagetoid, 1+, and 2+) the one-way ANOVA should be applied. The Student's t-test is restricted to comparison based on only 2 groups, not 3. Besides, how to compare statistically some values to the control group, whose value has no cases (with no pagetoid spread)?

Response 2: We appreciate the reviewer’s suggestion regarding statistical analysis. As recommended, we re-analyzed the data using one-way ANOVA across the three subgroups (0, 1+, 2+), which confirmed significant overall differences (F = 4.47, p = 0.019). Post-hoc testing revealed a significant difference between the no pagetoid spread group (0) and the 2+ group (p = 0.013), whereas the comparisons between 0 vs. 1+ (p = 0.281) and 1+ vs. 2+ (p = 0.125) were not statistically significant. (page 6, lines 207-213)

Comment 3:  Please, correct p-value statistics on the Figure 3. The decimal system in English is based on a period, not a comma.

Response 3: All p-values in the figures and throughout the manuscript have been corrected to follow the English decimal system, using a period instead of a comma.

Comment 4: Table 3 is very important as future directions and needs more explanation in the Discussion.

Response 4: We thank the reviewer for highlighting the importance of Table 3. We have expanded the Discussion to more clearly explain its relevance, emphasizing that the table provides a concise comparative overview of nested melanoma and melanocytic nevi, integrates our study results, and may serve as a practical reference for diagnostic decision-making. (page 11, lines 321-328)

Round 2

Reviewer 2 Report

Comments and Suggestions for Authors

The manuscript has been substantially improved.

Reviewer 3 Report

Comments and Suggestions for Authors

No further comments.